# Blood DNA Methylation in Nuclear and Mitochondrial Sequences Links to Malnutrition and Poor Prognosis in ALS: A Longitudinal Study

**DOI:** 10.3390/nu17081295

**Published:** 2025-04-08

**Authors:** Antia Fernandez-Pombo, Andrea G. Izquierdo, Ana Canton-Blanco, Tania Garcia-Sobrino, David Hervás, Miguel A. Martínez-Olmos, Julio Pardo, Ana B. Crujeiras

**Affiliations:** 1Epigenomics in Endocrinology and Nutrition Group, Epigenomics Unit, Instituto de Investigacion Sanitaria de Santiago de Compostela (IDIS), Complejo Hospitalario Universitario de Santiago de Compostela (CHUS), 15706 Santiago de Compostela, A Coruña, Spain; andrea.gonzalez.izquierdo@hotmail.com; 2Department of Endocrinology and Nutrition, Complejo Hospitalario Universitario de Santiago de Compostela (CHUS), University of Santiago de Compostela (USC), 15706 Santiago de Compostela, A Coruña, Spain; ana.canton.blanco@sergas.es (A.C.-B.); miguel.angel.martinez.olmos@sergas.es (M.A.M.-O.); 3CIBER Fisiopatologia de la Obesidad y Nutricion (CIBERobn), 28029 Madrid, Spain; 4Molecular Endocrinology Group, Instituto de Investigacion Sanitaria de Santiago de Compostela (IDIS), Complejo Hospitalario Universitario de Santiago de Compostela (CHUS), 15706 Santiago de Compostela, A Coruña, Spain; 5Department of Neurology, Complejo Hospitalario Universitario de Santiago (CHUS), 15706 Santiago de Compostela, A Coruña, Spain; tania.garcia.sobrino@sergas.es (T.G.-S.); julio.pardo.fernandez@sergas.es (J.P.); 6Department of Applied Statistics and Operational Research and Quality, Universitat Politècnica de València, 46022 Valencia, Valencia, Spain; daherma@upv.edu.es

**Keywords:** amyotrophic lateral sclerosis, Global Leadership Initiative on Malnutrition, epigenetics, malnutrition, D-loop, prognosis, blood leukocytes, mitoepigenetics

## Abstract

**Background**: Malnutrition in amyotrophic lateral sclerosis (ALS) is associated with disease severity, and epigenetic regulation may be involved. The aim of this study was to assess the methylation levels of specific DNA sequences from the nuclear and mitochondrial genomes in a population with ALS to elucidate their relationship with nutritional status and the evolution of the disease. **Methods**: Patients with ALS were evaluated between 2013 and 2021 (*n* = 66). They were categorized according to their nutritional status, using the Global Leadership Initiative on Malnutrition (GLIM) criteria, and disease progression, using the ALS Functional Rating (ALSFRS-R) Scale. DNA samples were extracted from leukocytes at the time of diagnosis for analysis of DNA methylation levels of markers of oxidative stress, mitochondrial function and global methylation (D-loop, GSTP1, and LINE-1). **Results**: According to the GLIM criteria, 29 (43.9%) patients had malnutrition (22.7%—moderate; 21.2%—severe), which was positively correlated with ALS disease progression (r = 0.414; *p* < 0.01) and death (r = 0.687; *p* < 0.01). Mortality occurred in 43.9% of the patients (median time to death, 18.7 (1.7–82.7) months). A significant association was observed between DNA methylation levels of the D-loop, GSTP1, and the CpG1 site of LINE-1 and malnutrition, disease progression at diagnosis, and death. The D-loop was the best predictor of malnutrition (AUC, 0.79; *p* < 0.01), disease progression (AUC, 0.70; *p* < 0.01), and mortality (AUC, 0.71; *p* < 0.01). **Conclusions**: This study revealed, for the first time, the early detection of D-loop methylation levels as a potential biomarker of nutritional status in patients with ALS, which may be useful for personalized nutritional management aimed at counteracting disease progression.

## 1. Introduction

Amyotrophic lateral sclerosis (ALS) is a neuromuscular disorder characterized by upper and lower motor neuron degeneration in the spinal cord, brainstem, and motor cortex, leading to progressive muscle atrophy and paralysis that can cause respiratory failure and death [1]. Its prevalence is approximately 5 cases per 100,000 individuals [2]. Genetic origin accounts for 47.7% of familial ALS cases and 5.2% of sporadic ALS cases [3]. Thus, ALS may be the outcome of multiple factors that, in addition to genetics, include physiological and environmental factors that contribute to the phenotypic variability reported in both sporadic and familial cases [4]. Although it is considered a severe and devastating disease, with an average life expectancy of 20–48 months from diagnosis, approximately 10% of patients survive more than 10 years [5,6].

Growing attention has been focused on the role of nutrition in the pathogenesis and prognosis of ALS and other neurodegenerative diseases [5,7,8,9,10]. Malnutrition is a major concern in patients with amyotrophic lateral sclerosis (ALS) due to anorexia, gastrointestinal disorders, and variations in energy expenditure. It is also an independent predictor of quality of life and survival [7,8]. In addition to worsening nutritional status, dysphagia promotes the development of respiratory infections [9]. Adequate nutritional support has been reported to have a positive impact on the progression of ALS [11,12].

Some medications, such as riluzole and edaravone, are currently used to attenuate symptoms [13]. However, there is still no definitive treatment for this disorder [14]. The phenotypic heterogeneity of ALS suggests the presence of mechanisms that may alter the characteristics of the disease [15]. Thus, the identification of these mechanisms could be of particular relevance in providing new targets and therapeutic strategies and in carrying out a more personalized nutritional plan.

Epigenetic regulation is a molecular mechanism by which environmental factors and lifestyle affect the functioning of an organism. Epigenetic mechanisms include DNA methylation marks, histone modifications, non-coding RNA (microRNA, long non-coding RNA) and sirtuin expression, epigenetic clock, mitochondrial DNA methylation, and telomere length. These mechanisms are involved in the cell-specific control of gene expression without changing the DNA sequence [16]. Epigenetic mechanisms play a relevant role in physiological and pathological conditions by regulating gene expression. In this context, there has been growing interest in studying the associations between epigenetic variations and disease susceptibility. Alterations in epigenetic regulation lead to the disruption of normal signaling pathways and may serve as relevant biomarkers for the progression of several disorders, including neurodegenerative diseases such as Alzheimer’s disease, Parkinson’s disease, and ALS [17,18,19]. In this regard, epigenetic regulation of mitochondrial function and control of oxidative stress are relevant processes involved in the pathogenesis of neurodegenerative diseases [20,21]. Previous studies have shown an association between the methylation levels of genes related to both nuclear and mitochondrial genomes, such as GSTP1 and LINE-1 [22,23] or the D-Loop and copy number variation [18], and ALS pathogenesis.

A relevant aspect is that the identification of DNA methylation markers can be useful for predicting the success of nutritional interventions [24], and when they are detected in blood leukocytes, they can be followed up in a longitudinal setting, even though epigenetic regulation is tissue-specific. This is particularly relevant when the target tissue of a disease, such as the brain in ALS, is difficult to access. Thus, the recognition of these epigenetic markers in blood leukocytes and the elucidation of their relationship with nutritional parameters and prognosis in ALS may constitute a potential tool for designing and prescribing personalized nutritional therapeutic strategies.

This study aimed to assess the methylation levels of specific DNA sequences from the nuclear and mitochondrial genomes related to the regulation of oxidative stress processes, mitochondrial function, and global methylation, such as GSTP1, D-loop, and LINE-1, in a population with ALS and to elucidate the relationship of these epigenetic mechanisms with the parameters of nutritional status and the progression of the disease.

## 2. Materials and Methods

### 2.1. Study Design and Population

In this retrospective, observational, longitudinal study, 66 patients diagnosed with ALS were evaluated from 2013 to 2021 in a specific ALS Unit at the University Clinical Hospital of Santiago de Compostela, Spain. This unit attends to patients from the healthcare areas of Santiago de Compostela and Barbanza (Galicia, Spain), with a total population of approximately 450,000 [25]. A multidisciplinary team of professionals from the Nutrition, Neurology, Pulmonology, Rehabilitation, Otolaryngology, and Psychology Departments was involved.

ALS was diagnosed using the revised El Escorial criteria [26]. Disease progression was determined at the time of diagnosis using the Revised Amyotrophic Lateral Sclerosis Functional Rating Scale (ALSFRS-R), establishing three prognostic subgroups while taking into account the rate of disease progression (∆FS) score [27], as follows: ∆FS < 0.47 indicates slow progression, ∆FS between 0.47 and 1.11 indicates moderate progression, and ∆FS > 1.11 indicates fast progression.

Nutritional status was evaluated at diagnosis in all patients, and nutritional data (development of dysphagia and need for enteral/parenteral nutrition) and clinical data (need for invasive mechanical ventilation, hospitalization, and death) were collected during follow-up. In addition, blood samples were extracted at the time of diagnosis from all patients for subsequent DNA isolation and analysis of DNA methylation levels of selected markers [27,28,29,30] and their correlation with the clinical and nutritional variables of the disease and its progression.

### 2.2. Nutritional Assessment and Support

Nutritional status was assessed in all patients at the first consultation in the Nutrition Department at the time of ALS diagnosis. It was performed using the Global Leadership Initiative on Malnutrition (GLIM) criteria [3,31] without considering muscle mass, taking into account the trophic alterations typical of the disease’s gradual muscle denervation. Thus, involuntary weight loss and/or low body mass index were considered as phenotypic criteria and were used to classify patients according to the severity of malnutrition. All the etiological criteria specified in this classification were taken into account [32]. In most cases, these criteria were met using the necessary data from the patients’ electronic medical records retrospectively, considering the 2019 consensus report [32]. Nutritional status was assessed in all patients by the same trained examiner.

Body weight and height were measured using a wall-mounted stadiometer adapted for wheelchairs (Seca 360° scale; Medical Resources, EPI Inc., Birmingham, AL, USA). The weight of the patients with mobility difficulties was obtained after discounting the previously known weight of the wheelchair. In these cases, the height of the patients was obtained from their most recent electronic clinical record.

Dysphagia was screened and diagnosed throughout the follow-up period using the Eating Assessment Tool 10 and the Volume–Viscosity Swallow Test.

To maintain and/or improve the patients’ energy and protein intake, nutritional support was prescribed according to the standard clinical practice for patients who did not meet their calculated needs. The type and amount of support were adjusted according to the patient’s situation, food intake, and presence of dysphagia. When necessary, enteral nutritional support was provided via nasogastric or gastrostomy tubes. After diagnosis, all the patients participated in a rehabilitation program that included respiratory physiotherapy and aerobic and resistance exercises.

### 2.3. DNA Methylation Analysis

Of the 66 patients in the overall sample, three patients were excluded from the DNA methylation analysis because some clinical data were not available. Figure 1 shows a flowchart of patient inclusion in this specific analysis and classification.

#### 2.3.1. DNA Preparation and Bisulfite Conversion

Genomic DNA was isolated from frozen whole blood samples using a MasterPure^TM^ DNA purification kit (Epicentre Biotechnologies, Madison, WI, USA) according to the manufacturer’s instructions. All the DNA samples were quantified using the fluorometric method (Qubit^TM^ 3 Fluorometer, dsDNA assays; ThermoScientific, Waltham, MA, USA) and assessed for purity using a NanoDrop (ThermoScientific, Waltham, MA, USA) to determine the 260/280 and 260/230 ratio measurements. The integrity of DNA was verified by electrophoresis on a 1.5% agarose gel. The DNA samples (500 ng) were bisulfite-converted using an EZ DNA Methylation Kit (Zymo Research, Tustin, CA, USA) according to the manufacturer’s instructions, which converts unmethylated cytosine into uracil.

#### 2.3.2. Pyrosequencing Analysis

Methylation analysis by pyrosequencing was performed on four CpG sites within the D-loop sequence (GenBank Accession No. J01415.2) [28], two CpG sites within the GSTP1 sequence (GenBank Accession No. M24485) [29], and four CpG sites within the LINE-1 sequence (GenBank Accession No. X58075) [30] in 63 patients with ALS. The primer sequences used in this analysis were designed using QIAGEN’s PyroMark Assay Design software (v. 2.0.2) (QIAGEN, Germantown, MD, USA) to hybridize CpG-free sites and ensure methylation-independent amplification (details and primer sequences are available in Appendix A). DNA methylation analyses were performed on bisulfite-treated DNA, followed by highly quantitative analysis using polymerase chain reaction (PCR)-based pyrosequencing with PyroMark Q24 system version 2.0.7 (QIAGEN, Hilden, Germany). Methylation levels were expressed as the percentage of methylated cytosines relative to the sum of methylated and unmethylated cytosines. Non-CpG cytosine residues were used as built-in controls to verify the bisulfite conversion. Values were expressed as the means for all sites. Human unmethylated (0% methylated DNA) and fully methylated DNA (100% methylated DNA) was also included as negative and positive controls in each run (Zymo Research). The reproducibility of all the samples was tested, and the failed assays were repeated. The inter-assay precision (%CV) was <2.5%, and intra-assay precision (%CV) was <1.0%.

### 2.4. Assessment of the Mitochondrial DNA Copy Number and Telomere Length

Absolute telomere length (TL) and mitochondrial DNA copy number (mtDNAcn) were quantified using a commercially available quantitative PCR (qPCR) assay kit (Absolute Human Telomere Length and Mitochondrial DNA Copy Number Dual Quantification qPCR Assay Kit, ScienCell Research Laboratories, #8958, Carlsbad, CA, USA) following the manufacturer’s protocol. The qPCR was conducted using a StepOnePlus Real-Time PCR System (Applied Biosystems, Thermo Fisher Scientific, Waltham, MA, USA). All the samples were tested in triplicate. After the qPCR was finished, absolute TL and mtDNAcn were calculated by using the ∆∆Ct method, where a single-copy reference primer set was used for data normalization.

### 2.5. Statistical Analysis

Data were expressed as the mean and the standard deviation (SD) or the mean ± standard error of the mean (SEM), the median and the interquartile range (IQR; 1st, 3rd quartile), or as percentages where applicable. The sample size was calculated to detect differences in methylation ≥10% with a type 1 error rate α = 0.05 and a power (1 − β) of 80%. The normal distribution of variables was explored using the Kolmogorov–Smirnov and Shapiro–Wilk tests. To study the differences between the groups, non-parametric tests (Kruskal–Wallis and Mann–Whitney tests) or parametric tests (Student’s *t*-test or analysis of variance) were used where applicable. The Holm–Sidak method was used to correct for multiple comparisons. The possible association between the DNA methylation levels analyzed and the parameters studied was evaluated using Spearman’s rho test. The ability of the selected biomarkers to discriminate among malnutrition, disease progression, and death was explored using simple logistic regression and receiver operating characteristic (ROC) curve analysis. The area under the ROC curve (AUC) was used to determine which biomarker was the best predictor of the evaluated clinical parameters. The level of significance was set at *p* < 0.05. All statistical analyses were performed using SPSS version 22.0 (Chicago, IL, USA) and GraphPad Prism version 9.5.1 for Windows (Boston, MA, USA).

## 3. Results

A total of 66 patients with a diagnosis of ALS (53.0%—men, mean age, 66.5 ± 10.1 years) were evaluated. The time from diagnosis to their first nutritional evaluation was 4.2 (0.0–18.6) months, and the median follow-up was 35.2 (12.2–87.7) months.

### 3.1. Clinical and Nutritional Evaluation

Table 1 shows the demographic, clinical, and nutritional status data at the first visit as well as the nutritional and clinical data during the follow-up of the overall sample (*n* = 66).

#### 3.1.1. Clinical and Nutritional Data at Diagnosis

While 59 patients (89.4%) presented sporadic ALS, only seven (10.6%) cases were familial (with pathogenic variants in the ALS2 and C9orf72 genes). In addition, ALS was classified as spinal-onset in 53 patients (80.3%) and as bulbar-onset in 13 patients (19.7%). Furthermore, according to the ALSFRS-R and ∆FS scores at diagnosis, 35 cases (53.0%) were classified as slow progression of the disease, 15 (22.7%)—as moderate progression, and 13 (19.7%)—as fast progression. Riluzole was administered to 63 patients (95.4%). Regarding the evaluation of nutritional status at the first visit (Table 1), the mean body mass index was 26.9 ± 4.1 kg/m^2^, and the percentage of weight loss was 9.8 ± 8.6%, with the median weight loss duration of 12.0 (3.0–24.0) months. According to the GLIM criteria, 29 patients (43.9%) had malnutrition. Of these, 15 cases (22.7%) were moderate and 14 (21.2%) were severe. Only five (7.6%) patients were being treated with folic acid and/or vitamin B (B1, B6, B12) supplementation at the moment of blood sampling for epigenomic analysis.

#### 3.1.2. Clinical and Nutritional Data During Follow-Up

During follow-up (Table 1), dysphagia was diagnosed in 36 cases (54.4%). Regarding nutritional support, 31 patients (46.9%) required oral nutritional supplements (with texture modification if needed), and although enteral nutrition via gastrostomy tube was recommended for 30 patients throughout follow-up, only 14 (21.2%) accepted it. In four cases, the administration of enteral nutrition through a nasogastric tube was necessary before the placement of the gastrostomy tube. In addition, one patient required ad hoc parenteral nutrition during hospital admission due to duodenal perforation. Fourteen patients (21.2%) required hospitalization for respiratory causes, and only one case accepted invasive mechanical ventilation. A total of 29 patients (43.9%) died during the follow-up period. The time to death since diagnosis was 18.7 (1.7–82.7) months.

### 3.2. Evaluation of DNA Methylation Levels in Relation to the Nutritional and Clinical Outcomes

When the DNA methylation levels of the selected markers (D-loop, GSTP1, and LINE-1) were evaluated, statistically significant differences were found in relation to malnutrition, disease progression, and death (Figure 2). The complete data are provided in Appendix A. No significant association was found between methylation levels and age, nor were the differences found based on sex; therefore, the analyses were not adjusted for these parameters. Lower methylation levels were observed in the D-loop of the patients with moderate and severe malnutrition according to the GLIM criteria (Figure 2A). In addition, the greater the severity of malnutrition, the lower the levels of methylation (1.34 ± 0.24% in the patients with severe malnutrition and 2.30 ± 1.39% in the patients without malnutrition; *p* < 0.001). On the other hand, regarding GSTP1 and LINE-1 at its CpG1 site, higher methylation levels were found in the patients with severe malnutrition, with GSTP1 showing a tendency toward greater differences the greater the severity of malnutrition (Figure 2B,C).

Similar results were observed when comparing DNA methylation levels in the D-loop with the progression of the disease and taking into account the ALSFRS-R and ∆FS scores (Figure 2D). Thus, patients considered to have fast disease progression showed significantly lower methylation levels (1.44 ± 0.33% in comparison with 2.19 ± 1.32% in individuals with slow progression; *p* = 0.004). Although a tendency toward higher methylation levels in GSTP1 in patients with fast disease progression was also observed, no significant differences were found (Figure 2E). Accordingly, death was associated with decreased methylation levels of the D-loop and increased methylation levels of GSTP1 (Figure 2G,H). Regarding LINE-1, slightly higher methylation levels were observed only at the CpG1 site in patients with fast progression (Figure 2F,I).

To elucidate the functional role of the changes observed in methylation levels in different subgroups of patients with ALS, mtDNAcn was evaluated together with TL as a measure of health and aging status [33] (Figure 3). Although no significant differences were observed between mtDNAcn and malnutrition, higher levels of mtDNAcn were found, particularly in those with fast disease progression (312.31 ± 191.88 copies per diploid cell in comparison with 123.55 ± 161.65 copies per diploid cell in the patients with slow progression; *p* = 0.0159). However, no significant differences were observed in survival.

### 3.3. Diagnostic Accuracy of the Selected Epigenetic Biomarkers in Relation to Nutritional and Clinical Outcomes

Malnutrition was positively correlated with ALS disease progression (r = 0.414; *p* < 0.01) and death (r = 0.687; *p* < 0.01) (Figure 4). As expected according to the results previously mentioned, an inverse correlation was also observed between the mean D-loop methylation levels and malnutrition, disease progression, and death (r = −0.535, r = −0.405, and r = −0.374, respectively; *p* < 0.01). In addition, based on the AUC shown in Figure 5, the D-loop was the best predictor of malnutrition (AUC = 0.79; *p* < 0.0001), disease progression (AUC = 0.70; *p* = 0.0057), and mortality (AUC = 0.71; *p* = 0.0048). On the other hand, methylation levels of GSTP1 and LINE-1 were positively correlated with these three clinical parameters only when CpG1 and CpG2 were analyzed separately (data shown in Figure 4).

## 4. Discussion

To the best of our knowledge, this is the first study showing the relationship between DNA methylation of specific biomarkers of oxidative stress, neuroprotection, cognitive behavior (fundamentally the D-loop, but also GSTP1 and the CpG1 site of LINE-1), and nutritional status in accordance with disease progression at diagnosis and mortality in patients with ALS.

Although ALS is considered a serious neurodegenerative disorder resulting in a greatly reduced life expectancy, strikingly, survival of more than 10 years is observed in approximately 10% of cases. Therefore, it is of particular relevance to elucidate the elements that, when acted upon, can have a positive impact on the evolution of the disease. Malnutrition, for example, is a factor to be considered. Its prevalence can vary between 9–16% when only the BMI is taken into account [34,35] and 48% when the GLIM criteria are applied [8]. This is consistent with our findings (44%, using the GLIM criteria), also showing a positive correlation with disease progression and death as reported in previous studies [11,35]. Adequate and early nutritional support may have a positive impact on the evolution of ALS [11,35]. These results underline the need for a multidisciplinary nutritional assessment, as supported by the ESPEN guidelines [32,36]. One of the causes that favor weight loss in patients with ALS is the development of dysphagia, which was diagnosed in 54.4% of cases in our cohort. Its prevalence in the different studies varies considerably, ranging between 6% and 75% depending on the evaluation method used [37].

Only 10% of ALS cases are considered to be familial [1], with more than 40 causative genes identified [38]. ALS is also characterized by a striking phenotypic heterogeneity, resulting in variations in disease evolution and prognosis [5,6]. These findings suggest the presence of other mechanisms that may affect the characteristics of this disorder [15]. For example, changes in epigenetic regulation, such as variations in DNA methylation in the peripheral blood, have been associated with the pathogenesis of neurodegenerative diseases [17,18,19]. Considering their reversibility [24], these changes may be used as markers and targets for the development of specific therapies that may change the course of the disease. In addition, although the tissue specificity of methylation can present a challenge, considering that the blood cell methylation profile may not necessarily reflect the epigenetic states in other tissues, several recent studies have shown that peripheral blood can mirror difficult-to-access target tissues [39,40], such as the brain. Accordingly, in blood cells, it has been demonstrated that D-loop methylation levels are significantly lower in individuals with neurodegenerative disorders [28], the methylation of GSTP1 is associated with susceptibility to DNA damage and increased cancer incidence [29], and the methylation status of LINE-1 is widely used as a surrogate marker of global DNA methylation, affecting the risk of cancer and cardiovascular and neurodegenerative diseases [41,42,43,44].

Regarding the D-loop, which is considered the central regulatory site for the replication and transcription processes of mitochondrial DNA, methylation levels have been observed to be lower in sporadic ALS cases than in healthy controls [28], as well as in ALS patients with C9orf72 repeat expansions and SOD1 variants [18]. Only one recent report suggested an inverse correlation between D-loop methylation levels and disease duration and a positive correlation between the latter and mtDNAcn [18]. In the present study, these results were corroborated, and an association was also observed between lower D-loop methylation levels and greater disease progression and mortality, along with malnutrition at diagnosis. It is also known that variants in mitochondrial DNA may lead to an increase in the formation of cellular reactive oxygen species in neurodegenerative disorders [45]. Taking all of this into account, the D-loop seems to be an indicator of a worse evolution of the disease in ALS, and, according to our results, we suggest that nutritional status could influence the variations in this biomarker. This could open the door for the development of specific therapies to counteract mitochondrial impairment and oxidative stress in ALS, with the aim of reversing these epigenetic changes. In this sense, the possibility that diets that improve mitochondrial function may also improve the prognosis of ALS has been explored in several studies. Specifically, the Mediterranean diet has demonstrated its impact on reducing oxidative stress and inflammation. Supplementation with antioxidants such as nicotinamide riboside and pterostilbene and the use of coconut oil as an alternative energy source have shown anthropometric benefits in patients with ALS, favoring the preservation of muscle mass [46]. The ketogenesis derived from this type of diet has been observed to have a neuroprotective effect, optimizing the energy balance in affected motor neurons [47]. Other studies have indicated that certain phytochemical compounds present in this diet could stimulate mitophagy, promoting the removal of dysfunctional mitochondria and therefore reducing oxidative stress and inflammation in ALS [48]. Nutritional strategies aimed at modulating the gut microbiome have also been proposed, as intestinal dysbiosis is linked to neuroinflammation and oxidative stress in ALS [49]. Bioactive compounds, such as resveratrol, have shown an improvement in mitochondrial function by promoting the activity of SIRT1 and PGC-1α [50]. The impact of CoQ10 deficiency on mitochondrial function, oxidative stress, and certain neurodegenerative diseases has also been described [51]. In addition, further investigation of certain immunonutrients and other bioactive compounds with antioxidant properties, such as EPA and DHA (which participate in the modulation of systemic inflammation) and beta-glucans (which present immunoregulatory functions that favor the reduction of proinflammatory cytokine levels) [52], would be of interest.

Higher methylation levels of LINE-1 at its CpG1 site and GSTP1 were also found in the patients with severe malnutrition. However, these changes, even though statistically significant, were not as pronounced as those observed in the D-Loop. This could be explained by the specific functions of the markers analyzed in disease regulation. The D-loop, as a control region of mitochondrial DNA, is highly sensitive to metabolic and oxidative stress, which are key factors in malnutrition and ALS progression. In contrast, GSTP1 and LINE-1 may be regulated by additional factors that were not captured in our study, such as specific environmental exposures or other disease-related epigenetic modifications.

GSTP1 may also alter the effectiveness of the products of oxidative stress and neurotoxins, contributing to cell protection and detoxification [53]. In fact, it has been suggested that GSTP1 polymorphisms may also play a role in excessive oxidative stress and the promotion of motor neuron death involved in neurodegenerative disorders such as Alzheimer’s disease and Parkinson’s disease [53,54,55]. Although no relationship has been found between GSTP1 polymorphisms or GSTP1 variant carriers and an increased risk of ALS [54,56], lower survival rates have been observed in male carriers of this variant [57]. This is the first study showing differences in the methylation levels of GSTP1 in ALS. Higher methylation levels in this biomarker were associated with severe malnutrition and death, and a tendency toward higher methylation was observed in the patients with fast disease progression. Although it was not possible to obtain samples to evaluate the effects of these differences on gene expression, high methylation levels of GSTP1 were previously associated with lower expression under exposure to benzopyrene [58] or arsenic [59].

Apart from oxidative stress, retrotransposition has also been correlated with neurodegenerative diseases such as ALS or cognitive dysfunction in the form of frontotemporal dementia, present in 20–25% of ALS cases [36]. In fact, the loss of nuclear TDP-43 has been associated with the decondensation of LINE retrotransposons in postmortem frontotemporal degeneration-ALS human brains [60]. LINE-1 activity contributes to genome instability and is considered a surrogate marker for global DNA methylation [61]. Reduced levels of LINE-1 methylation in the motor cortex of both familial and sporadic ALS cases, compared to control brains, were found in a previous study [62]. However, in our study, greater methylation levels were found at position 1 in relation to severe malnutrition and rapid disease progression. It should be noted that the previous analysis included only a small number of highly active LINE-1 loci [62]. In another study conducted in women with no neurodegenerative diseases, LINE-1 methylation tended to decrease from underweight women to women with obesity [63]. Thus, further studies are required to fully explore the potential role of LINE-1 methylation in the predisposition to ALS and its impact on nutritional derangement and prognosis. For now, what is known is that deficits in neurogenic function and cognitive or affective behaviors may be modulated by both specific diets and environmental enrichment and that some of these behaviors might be indirectly associated with LINE-1 methylation [30].

Taken together, according to the findings of the current study, early detection of these epigenetic and reversible biomarkers (specifically the D-loop) in patients with ALS may not only be considered a motor neuron degeneration factor, as previously shown in the literature, but, taking into account their relationship with malnutrition and disease progression, may also be used as a target for the development of personalized nutritional therapies that might have clinical implications and contribute to changing the course of the disease. Although a limitation of this study was the small sample size, the patient cohort was sufficiently homogeneous to avoid possible bias in the analyses, and the changes detected in the methylation profiles reported differences with statistical power. The sample size and data obtained can also be considered representative of the general ALS population considering the prevalence of the disorder and disease subtypes (according to ALS origin and onset), progression [1,2,64] and taking into account that all patients were evaluated at the Nutrition Department regardless of their nutritional status; therefore, the risk of selecting only patients with a worse nutritional status was avoided. Methylation analysis was carried out in whole blood, which includes cellular heterogeneity and may have reduced the capacity to detect further differences in DNA methylation levels. Blood samples were collected from a biobank where they were not preserved under optimal conditions to obtain high-quality RNA and evaluate the functionality of methylation levels of the GSTP1 promoter. Moreover, as previously mentioned, it is known that certain nutrients may affect epigenetic changes such as DNA methylation [46,47,48,49,50,51,52]. Unfortunately, the patients included in the study did not complete a food intake or physical activity questionnaire at diagnosis. Another limitation of the present study was the lack of data related to muscle mass during nutritional assessment. However, trophic alterations related to the muscle denervation characteristics of ALS make it difficult to accurately differentiate ALS from muscle wasting in the context of malnutrition, even for expert evaluators. Therefore, other phenotypic criteria were used to evaluate the GLIM criteria. On the other hand, it is worth highlighting the presence of the same trained examiner when assessing the nutritional status of all the patients evaluated, which confers greater consistency in the data presented.

Future research in this field should focus on validating these biomarkers through multicenter longitudinal studies, which would allow a deeper understanding of their mechanisms of action in relation to nutritional status and disease outcomes. Additionally, investigating the potential therapeutic effects of targeted nutritional interventions based on these specific epigenetic profiles will be crucial. In this context, emerging technologies could also play a transformative role. The Internet of Things may enable continuous monitoring of nutritional parameters and nutritional management in patients with chronic neurological diseases [65], facilitating real-time adjustments tailored to individual needs. Likewise, artificial intelligence holds great promise for developing predictive models to identify at-risk patients, optimize dietary interventions, and personalize nutritional strategies through large-scale data analysis [66,67]. These innovations, together with advances in omics approaches, could contribute to improving the progression of the disease in these patients.

## 5. Conclusions

Although a relationship between certain epigenetic markers and the development of neurodegenerative diseases such as ALS has already been proposed, this study is the first to relate the D-loop, GSTP1, and the CpG1 site of LINE-1 with worse nutritional status in accordance with disease progression at diagnosis and mortality, with the D-loop being the best predictor for these outcomes. Early identification of these epigenetic markers may enable early personalized nutritional management, with the aim of reversing epigenomic modifications and improving patient survival (Figure 6). Multicenter longitudinal studies are needed to validate these biomarkers, explore their mechanisms of action in relation to nutritional status and disease outcomes, and investigate the possible effects and therapeutic roles of targeted nutritional interventions based on these specific epigenetic profiles.

## Figures and Tables

**Figure 1 nutrients-17-01295-f001:**
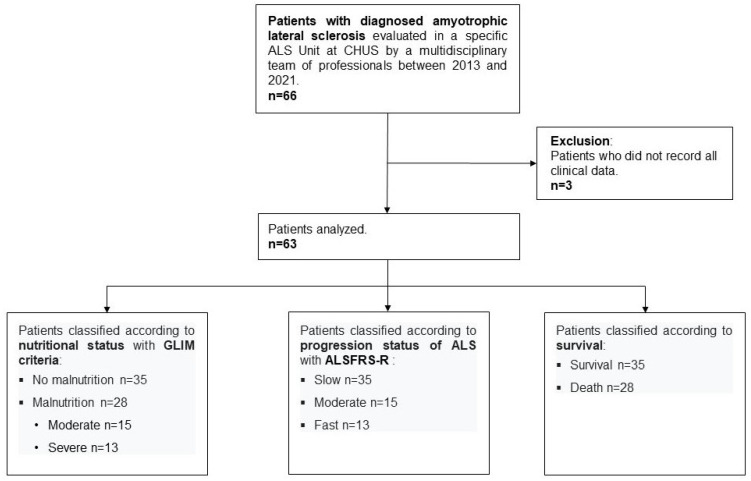
Flowchart indicating the inclusion of patients for DNA methylation analysis. ALS, amyotrophic lateral sclerosis; CHUS, Complejo Hospitalario Universitario de Santiago; GLIM, Global Leadership Initiative on Malnutrition; ALSFRS-R, Revised Amyotrophic Lateral Sclerosis Functional Rating Scale.

**Figure 2 nutrients-17-01295-f002:**
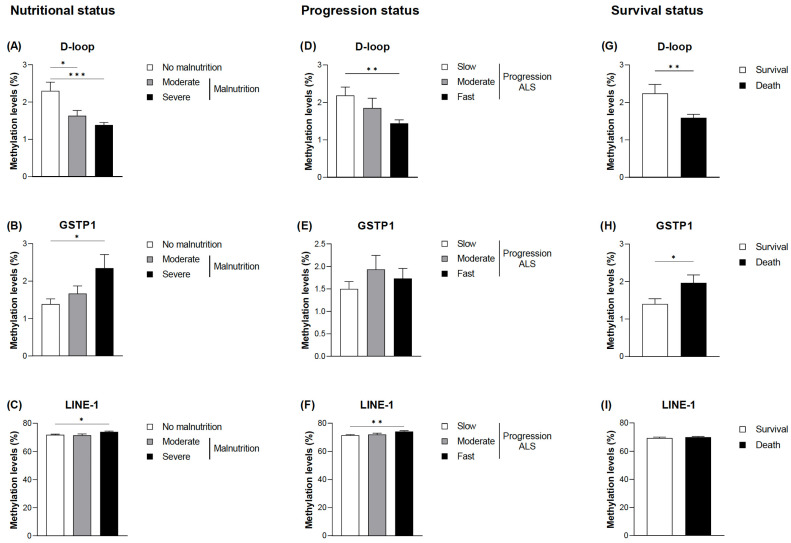
DNA methylation based on the nutritional status and the progression status at diagnosis of ALS and mortality. Differences in the mean DNA methylation levels at the D-loop CpG sites (**A**), GSTP1 CpG sites (**B**), and LINE-1 CpG1 sites (**C**) based on the nutritional status according to the GLIM criteria. Differences in the mean DNA methylation levels at the D-loop CpG sites (**D**), GSTP1 CpG sites (**E**), and LINE-1 CpG1 sites (**F**) based on the progression status according to ALSFRS-R. DNA methylation levels at the D-loop CpG sites (**G**), GSTP1 CpG sites (**H**), and LINE-1 CpG1 sites (**I**) based on survival. The data are presented as the mean ± SEM values. Statistically significant differences: * *p* < 0.05, ** *p* < 0.01, and *** *p* < 0.001. GLIM, Global Leadership Initiative on Malnutrition; ALSFRS-R, Revised Amyotrophic Lateral Sclerosis Functional Rating Scale; ALS, amyotrophic lateral sclerosis.

**Figure 3 nutrients-17-01295-f003:**
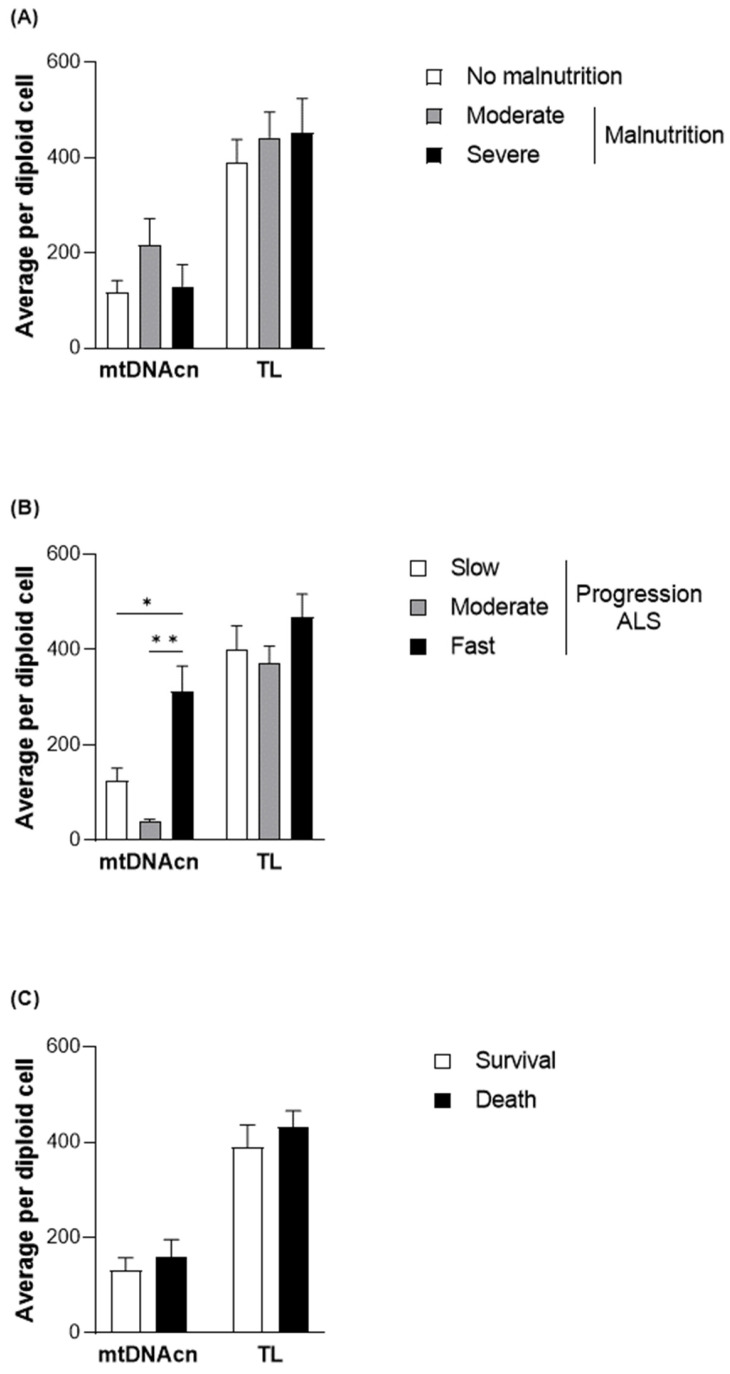
TL and mtDNAcn quantified in DNA samples from the patients with ALS at diagnosis. Differences in the average quantification per diploid cell of TL and mtDNAcn based on nutritional status according to the GLIM criteria (**A**), progression status according to the ALSFRS-R (**B**), and survival (**C**) of patients with ALS. Data are presented as the mean ± SEM values. Statistically significant differences: * *p* < 0.05, ** *p* < 0.001. GLIM, Global Leadership Initiative on Malnutrition; ALSFRS-R, Revised Amyotrophic Lateral Sclerosis Functional Rating Scale; mtDNAcn, mitochondrial DNA copy number; TL, absolute telomere length; ALS, amyotrophic lateral sclerosis.

**Figure 4 nutrients-17-01295-f004:**
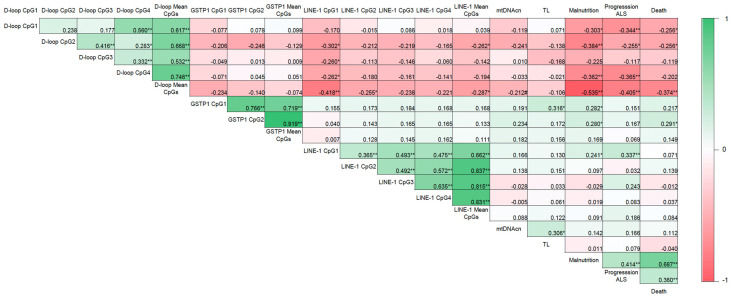
Correlogram showing the relationship between DNA methylation and the clinical parameters evaluated. Correlation matrix between DNA methylation and malnutrition (according to the GLIM criteria), ALS progression (according to the ALSFRS-R), and death in patients with ALS. Correlation coefficients were evaluated using Pearson’s and Spearman’s tests. Note: * *p* < 0.05, ** *p* < 0.01, # *p* = 0.099; GLIM, Global Leadership Initiative on Malnutrition; ALSFRS-R, Revised Amyotrophic Lateral Sclerosis Functional Rating Scale; mtDNAcn, mitochondrial DNA copy number; TL, absolute telomere length; ALS, amyotrophic lateral sclerosis.

**Figure 5 nutrients-17-01295-f005:**
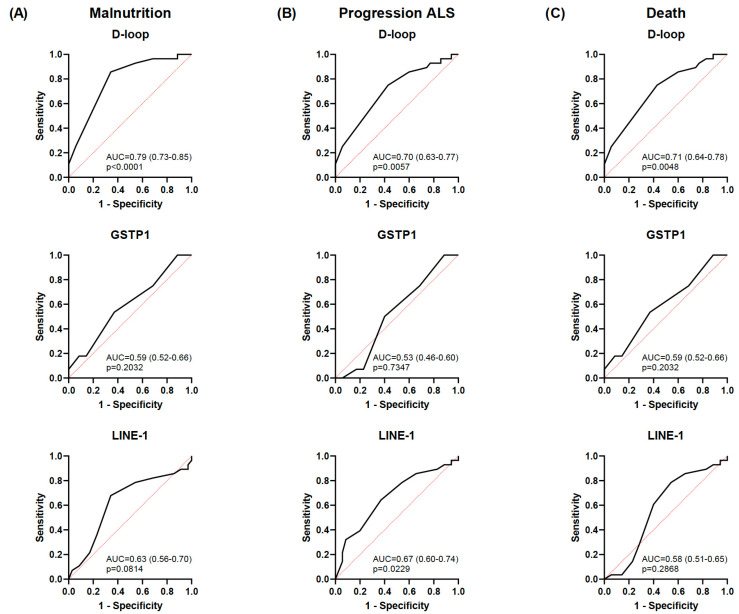
ROC curves for each biomarker for predicting malnutrition, disease progression, and death. ROC curves showing the ability to discriminate malnutrition (GLIM criteria) (**A**), progression of ALS (ALSFRS-R) (**B**), and death of patients with ALS (**C**) for the D-loop, GSTP1, and LINE-1. Data are presented as the mean ± SD values. The black line represents the simple logistic regression curve, and the red line is the identity or reference line. ROC, receiver operating characteristic; ALS, amyotrophic lateral sclerosis; GLIM, Global Leadership Initiative on Malnutrition; ALSFRS-R, Revised Amyotrophic Lateral Sclerosis Functional Rating Scale.

**Figure 6 nutrients-17-01295-f006:**
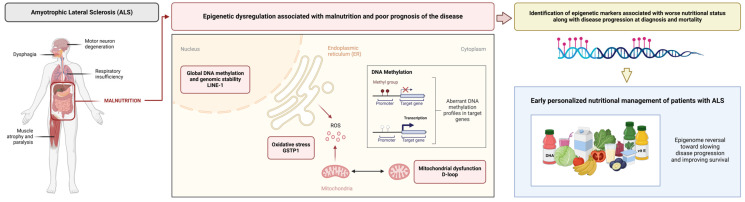
Epigenetic markers related to oxidative stress, mitochondrial function, and global DNA methylation, such as the D-loop, GSTP1, and LINE-1, are associated with malnutrition along with poor prognosis in ALS. Identification of these markers may enable early personalized nutritional therapy in order to reverse these epigenomic modifications and improve outcomes.

**Table 1 nutrients-17-01295-t001:** Demographic, anthropometric, and nutritional status data at the first visit and at follow-up of patients with amyotrophic lateral sclerosis.

**Data at diagnosis of ALS**	
Age, years	66.5 ± 10.1
Gender, *n* (%) (men/women)	35 (53.0)/31 (47.0)
ALS origin	
Sporadic, *n* (%)	59 (89.4)
Familial, *n* (%)	7 (10.6)
ALS onset	
Spinal, *n* (%)	53 (80.3)
Bulbar, *n* (%)	13 (19.7)
Weight, kg	70.0 ± 12.5
BMI, kg/m^2^	26.9 ± 4.1
Change of BW, %	−4.77 ± 9.3
Weight loss, %	9.82 ± 8.6
Nutritional assessment (GLIM)	
No malnutrition, *n* (%)	37 (56.1)
Moderate malnutrition, *n* (%)	15 (22.7)
Severe malnutrition, *n* (%)	14 (21.2)
ALSFRS-R	
Slow progression, *n* (%)	35 (53.0)
Moderate progression, *n* (%)	15 (22.7)
Fast progression, *n* (%)	13 (19.7)
**Data during follow-up**	
Nutritional support	
ONS, *n* (%)	31 (46.9)
Gastrostomy, *n* (%)	14 (21.2)
Dysphagia, *n* (%)	36 (54.4)
Hospitalization, *n* (%)	14 (21.2)
Death, *n* (%)	29 (43.9)

The data (*n* (%)) are presented as the mean ± SD or median (IQR) values. ALS, amyotrophic lateral sclerosis; BMI, body mass index; GLIM, Global Leadership Initiative on Malnutrition; ALSFRS-R, Revised Amyotrophic Lateral Sclerosis Functional Rating Scale; ONS, oral nutritional supplements.

## Data Availability

The data presented in this study are available upon request from the corresponding authors due to privacy and ethical restrictions.

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
