# Peer review of "Blood DNA Methylation in Nuclear and Mitochondrial Sequences Links to Malnutrition and Poor Prognosis in ALS: A Longitudinal Study"

_nutrients, 2025, doi:10.3390/nu17081295_

Round 1

Reviewer 1 Report

Comments and Suggestions for Authors

This study comprehensively evaluated DNA methylation, which affects the nutritional status of amyotrophic sclerosis. The main results of the study are as follows: The authors showed that there is a close association between the nutritional status and DNA methylation of ALS patients. In particular, the authors suggest that the methylation level of the D-loop region of mitochondrial DNA may be a useful biomarker for predicting nutritional status, disease progression, and death.

#1Regarding mitochondrial function, the authors' description in the discussion section is difficult to understand, as to whether they think it is related to ALS disease or nutritional status. In other words, it would be good to state whether the D-loop is considered to be a trait marker or a state marker of ALS.

#2 We invite further consideration of what can be said about the fact that a relationship between GSTP1, LINE-1 and malnutrition in ALS was not observed, as was the case for the D-loop.

#3 Nutritional management is important in the treatment of ALS. This discovery may open new avenues for improving nutritional management of ALS patients and slowing disease progression. It is helpful to review the literature on the possibility that diets that improve mitochondrial function may improve the prognosis of ALS.

#4 Certain nutrients may affect epigenetic changes such as DNA methylation and histone modifications. For example, folic acid and B vitamins are involved in DNA methylation, and polyphenols contained in green and yellow vegetables and berries may affect histone modifications. Exercise has been reported to affect DNA methylation and histone modifications in muscle and brain tissues. It is important to know whether these factors are different in the context of this study. Please add any comments on this.

Author Response

This study comprehensively evaluated DNA methylation, which affects the nutritional status of amyotrophic sclerosis. The main results of the study are as follows: The authors showed that there is a close association between the nutritional status and DNA methylation of ALS patients. In particular, the authors suggest that the methylation level of the D-loop region of mitochondrial DNA may be a useful biomarker for predicting nutritional status, disease progression, and death.

RESPONSE: We would like to thank the reviewer for their expertise and time in reviewing our work and for providing these constructive comments and suggestions, which will contribute to improving this manuscript.

#1Regarding mitochondrial function, the authors' description in the discussion section is difficult to understand, as to whether they think it is related to ALS disease or nutritional status. In other words, it would be good to state whether the D-loop is considered to be a trait marker or a state marker of ALS.

RESPONSE 1. We agree with the reviewer that this paragraph, as written, can lead to confusion. Considering our results, as well as another recent report (PMID: 38312023), decreased D-loop methylation levels may contribute to disease progression and mortality in ALS and, considering the association of these variables and methylation levels with malnutrition in our study, we also propose that nutritional status could influence the changes in D-loop. Therefore, D-loop is considered to be a state marker of ALS rather than a trait. D-loop seems to be an indicator of a worse evolution of the disease in ALS and, according to our results, we suggest that nutritional status could influence the variations in this biomarker. This has now been better specified in the Discussion section (lines 374-379).

#2 We invite further consideration of what can be said about the fact that a relationship between GSTP1, LINE-1 and malnutrition in ALS was not observed, as was the case for the D-loop.

RESPONSE 2. Thank you for your insightful comment. In fact, a relationship between GSTP1, LINE-1, and malnutrition and prognosis in ALS was observed; however, the association was less pronounced compared to the D-loop. This suggests that while epigenetic modifications in GSTP1 and LINE-1 may contribute to malnutrition and progression in ALS, the D-loop appears to be more sensitive to metabolic stress. This difference could be due to the distinct regulatory mechanisms governing nuclear and mitochondrial DNA methylation in response to malnutrition.

In our study, we observed significant differences in methylation levels of GSTP1 and LINE-1 in relation to the nutritional status of ALS patients, specifically in relation to severe malnutrition (Figure 2B-C and Table S2). Furthermore, these methylation changes, mainly those observed at the CpG1 and CpG2 sites of the GSTP-1 gene and at the CpG1 site of LINE-1, were correlated with malnutrition (GSTP1 CpG1: r=0.282, p<0.05; GSTP1 CpG2: r=0.280, p<0.05; LINE-1 CpG1: r=0.241, p<0.05) (Figure 4). However, these changes were not as pronounced as those observed in D-Loop, so we agree that the relationship detected between D-Loop methylation and malnutrition in ALS was stronger (Figure 2A and Table S2), which led to its identification as the best predictor (Figure 5A).

We consider that these findings regarding malnutrition in ALS could be explained by the specific functions of the markers analyzed in disease regulation.

We have clarified this point in the revised manuscript, emphasizing the varying degrees of association observed (please see text marked in red).

#3 Nutritional management is important in the treatment of ALS. This discovery may open new avenues for improving nutritional management of ALS patients and slowing disease progression. It is helpful to review the literature on the possibility that diets that improve mitochondrial function may improve the prognosis of ALS.

RESPONSE 3. We really appreciate this comment from the reviewer and the opportunity to provide information regarding potential dietary patterns or specific dietary compounds that can be able to recover mitochondrial function and potentially improve the prognosis of ALS. Most of these dietary patterns could also have the capacity to modulate epigenetic mechanisms. This topic was included in the new version of the manuscript (lines 381-401 in the discussion section).

#4 Certain nutrients may affect epigenetic changes such as DNA methylation and histone modifications. For example, folic acid and B vitamins are involved in DNA methylation, and polyphenols contained in green and yellow vegetables and berries may affect histone modifications. Exercise has been reported to affect DNA methylation and histone modifications in muscle and brain tissues. It is important to know whether these factors are different in the context of this study. Please add any comments on this.

RESPONSE 4. Only five (7.6%) patients were being treated with folic acid and/or vitamin B (B1, B6, B12) supplementation at the moment of blood sampling for epigenomic analysis. After diagnosis, all the patients participated in a rehabilitation program that included respiratory physiotherapy and aerobic and resistance exercises. Unfortunately, the patients included in the study did not complete a food intake or physical activity questionnaire at diagnosis. In agreement with the reviewer, it would be interesting to have this information in relation to the observed changes in DNA methylation, and this limitation is now acknowledged in the study.

We have added this information in the Methods section (lines 145-147), Results section (lines 238-239) and a brief comment on this in the Discussion section (lines 459-462). Please see text marked in red. We appreciate this suggestion.

Reviewer 2 Report

Comments and Suggestions for Authors

Dear Authors,
Thank you for submitting your manuscript titled Blood DNA Methylation Profile in Specific Sequences from the Nuclear and Mitochondrial Genome at Diagnosis Associates with Malnutrition and Poor Prognosis in Amyotrophic Lateral Sclerosis. I find the topic highly relevant, and I am pleased to review your manuscript. Below are my comments and suggestions:

TITLE:
The title could be shortened, and it would be helpful to specify the study design.

INTRODUCTION:

  • I recommend adding a section at the beginning of the introduction that includes epidemiological data on ALS, followed by a discussion of malnutrition as a complication.
  • Lines 71-73: Please insert a citation.
  • I would expand the section discussing “Epigenetic Regulation” for a more detailed explanation.

METHODS:
The methods section is well-described and clear. I recommend structuring the study according to STROBE guidelines.

RESULTS:

  • In Table 1, you mention that 54.4% had dysphagia. Could you include the severity grade?
  • Figure 3 could be enlarged for better clarity.

I believe the results section demonstrates high scientific quality, and I commend the authors for their work.

DISCUSSION:

  • In the discussion section, when addressing malnutrition, it would be beneficial to also mention dysphagia. Specifically, your sample exhibited a high degree of dysphagia, as evidenced by the complications of this condition.
  • You could also expand this section by comparing your malnutrition data with other studies on neurodegenerative and neurological diseases. This would further emphasize the importance of a multidisciplinary nutritional assessment, as supported by the ESPEN guidelines, which you have correctly cited. In my opinion, these additions could strengthen your results.
  • Another important aspect is the future directions of your study. While you’ve correctly mentioned them in the conclusion, I suggest creating a brief paragraph at the end of the discussion. Given the innovative nature of your study, I believe future research could be enriched by emerging technologies. For instance, you might include to the internet of things in the nutritional management of patients with chronic neurological disease or the application of artificial intelligence technologies to nutritional management, as well as other new technologies.

I congratulate the authors on their innovative study and the quality of the manuscript.

REFERENCES:
The references should include the DOI of each study for proper citation.

Author Response

Dear Authors,

Thank you for submitting your manuscript titled Blood DNA Methylation Profile in Specific Sequences from the Nuclear and Mitochondrial Genome at Diagnosis Associates with Malnutrition and Poor Prognosis in Amyotrophic Lateral Sclerosis. I find the topic highly relevant, and I am pleased to review your manuscript. Below are my comments and suggestions:

RESPONSE: First of all, we would like to thank the reviewer for the constructive remarks and valuable suggestions, which will definitely contribute to improving this manuscript.

TITLE:

The title could be shortened, and it would be helpful to specify the study design.

RESPONSE: The title of the manuscript has been changed following the reviewer’s suggestion.

INTRODUCTION:

I recommend adding a section at the beginning of the introduction that includes epidemiological data on ALS, followed by a discussion of malnutrition as a complication.

RESPONSE: We agree with the reviewer that starting the Introduction section with a presentation of the disease and its epidemiological data provides an adequate context before moving on to refer to malnutrition. This part of the text has been modified accordingly (see text marked in red in the Introduction section of the manuscript).

Lines 71-73: Please insert a citation.

RESPONSE: These lines were not intended to be an affirmation. What we really wanted to express was the possibility of providing new therapies if the mechanisms that determine the heterogeneity of the disease could be identified. We understand that this can be confusing and, therefore, this sentence has been changed to a conditional. Please see text marked in red.

I would expand the section discussing “Epigenetic Regulation” for a more detailed explanation.

RESPONSE: Following the reviewer’s suggestion, a brief paragraph with further explanation regarding epigenetic regulation has been added to the Introduction section (see text marked in red).

METHODS:

The methods section is well-described and clear. I recommend structuring the study according to STROBE guidelines.

RESPONSE: The article is currently adapted to the STROBE guidelines. Some minor changes have been made, such as in the title, which now reflects the design of the study.

RESULTS:

In Table 1, you mention that 54.4% had dysphagia. Could you include the severity grade?

RESPONSE: Indeed, it would be of interest to have more information about the degree of severity of dysphagia in the patients evaluated. However, as mentioned in the Methods section (lines 139-140), dysphagia was screened and diagnosed using the Eating Assessment Tool 10 (EAT-10) and Volume-Viscosity Swallow Test (V-VST). These tests are used to assess the risk of presenting dysphagia (in the case of EAT-10) and offer a clinical diagnosis of dysphagia, also allowing the viscosity of thickeners to be adjusted if necessary (in the case of V-VST). Unfortunately, they are not used to classify dysphagia according to severity.

Figure 3 could be enlarged for better clarity.

RESPONSE: Following the reviewer's suggestion, we have enlarged the size of Figure 3 for better clarity.

I believe the results section demonstrates high scientific quality, and I commend the authors for their work.

RESPONSE: We greatly appreciate the reviewer’s positive comment.

DISCUSSION:

In the discussion section, when addressing malnutrition, it would be beneficial to also mention dysphagia. Specifically, your sample exhibited a high degree of dysphagia, as evidenced by the complications of this condition.

RESPONSE: Following the reviewer’s suggestion, a brief paragraph regarding dysphagia in this population has been added to the Discussion section (see lines 346-349)

You could also expand this section by comparing your malnutrition data with other studies on neurodegenerative and neurological diseases. This would further emphasize the importance of a multidisciplinary nutritional assessment, as supported by the ESPEN guidelines, which you have correctly cited. In my opinion, these additions could strengthen your results.

RESPONSE: As recommended, our malnutrition data was compared in more detail with other studies on ALS, also highlighting the relevance of a multidisciplinary nutritional assessment, as supported by the ESPEN guidelines. See lines 340-346 in the Discussion section.

Another important aspect is the future directions of your study. While you’ve correctly mentioned them in the conclusion, I suggest creating a brief paragraph at the end of the discussion. Given the innovative nature of your study, I believe future research could be enriched by emerging technologies. For instance, you might include to the internet of things in the nutritional management of patients with chronic neurological disease or the application of artificial intelligence technologies to nutritional management, as well as other new technologies.

RESPONSE: We thank the reviewer for this comment. We believe that the addition of this final paragraph contributes to giving a better ending to the article. Please, see text in red at the end of the Discussion section.

I congratulate the authors on their innovative study and the quality of the manuscript.

RESPONSE: We appreciate the kind words of the reviewer and, again, their efforts to improve this manuscript.

REFERENCES:

The references should include the DOI of each study for proper citation.

RESPONSE: We agree with the reviewer that the presence of identifiers, such as the DOI, would allow a better identification of the articles selected as references. However, we have followed the instructions of the journal regarding this matter. Thus, the references have been structured according to the specific journal guidelines.

Round 2

Reviewer 2 Report

Comments and Suggestions for Authors

Dear Authors,

I greatly appreciate the revisions made and believe the manuscript now demonstrates an adequate methodological quality.

I kindly request that in the final paragraph of the discussion, you support the integrated text with appropriate and recent references:

"In this context, emerging technologies could also play a transformative role. The Internet of Things may enable continuous monitoring of nutritional parameters in patients with ALS, facilitating real-time adjustments tailored to individual needs. Likewise, artificial intelligence holds great promise for developing predictive models to identify at-risk patients, optimize dietary interventions, and personalize nutritional strategies through large-scale data analysis. These innovations, together with advances in omics approaches, could contribute to improving the evolution of the disease in these patients."

Specifically, I ask you to better support the text regarding:

- the Internet of Things in the nutritional management of patients with chronic neurological diseases
- the application of artificial intelligence technologies to nutritional management

Author Response

Dear Authors,

I greatly appreciate the revisions made and believe the manuscript now demonstrates an adequate methodological quality.

I kindly request that in the final paragraph of the discussion, you support the integrated text with appropriate and recent references:

"In this context, emerging technologies could also play a transformative role. The Internet of Things may enable continuous monitoring of nutritional parameters in patients with ALS, facilitating real-time adjustments tailored to individual needs. Likewise, artificial intelligence holds great promise for developing predictive models to identify at-risk patients, optimize dietary interventions, and personalize nutritional strategies through large-scale data analysis. These innovations, together with advances in omics approaches, could contribute to improving the evolution of the disease in these patients."

Specifically, I ask you to better support the text regarding:

- the Internet of Things in the nutritional management of patients with chronic neurological diseases

- the application of artificial intelligence technologies to nutritional management

RESPONSE: We appreciate the reviewer’s feedback and valuable suggestion. We have now incorporated the appropriate and recent references to support the final paragraph of the Discussion section regarding the role of IoT and AI in nutritional management. Please, see the text marked in red.

Round 3

Reviewer 2 Report

Comments and Suggestions for Authors

congratulations to the authors for this manuscript